# Energy Consumption and CO$_2$ Emissions in Ironmaking and Development of a Novel Flash Technology

**Hong Yong Sohn** 

Department of Materials Science & Engineering, University of Utah, 135 South 1460 East, Salt Lake City, UT 84112, USA; h.y.sohn@utah.edu; Tel.: +1-801-581-5491

**Abstract:** The issues of energy consumption and CO$_2$ emissions of major ironmaking processes, including several new technologies, are assessed. These two issues are interconnected in that the production and use of fuels to generate energy add to the total amount of CO$_2$ emissions and the efforts to sequester or convert CO$_2$ require energy. The amounts of emissions and energy consumption in alternate ironmaking processes are compared with those for the blast furnace, currently the dominant ironmaking process. Although more than 90% of iron production is currently through the blast furnace, intense efforts are devoted to developing alternative technologies. Recent developments in alternate ironmaking processes, which are largely driven by the needs to decrease CO$_2$ emissions and energy consumption, are discussed in this article. This discussion will include the description of the recently developed novel flash ironmaking technology. This technology bypasses the cokemaking and pelletization/sintering steps, which are pollution prone and energy intensive, by using iron ore concentrate. This transformational technology renders large energy saving and decreased CO$_2$ emissions compared with the blast furnace process. Economic analysis indicated that this new technology, when operated using natural gas, would be economically feasible. As a related topic, we will also discuss different methods for computing process energy and total energy requirements in ironmaking.

**Keywords:** ironmaking; carbon emissions; energy consumption; flash ironmaking process; alternate ironmaking processes; direct reduction; smelting reduction; iron ore concentrate; natural gas

## 1. Introduction

The blast furnace (BF), direct reduction (DR), and recently developed smelting reduction (SR) make up major ironmaking processes currently practiced in industry. The BF process currently produces more than 90% of the world production [1]. The balance is produced largely by gas-based direct reduction, with the rotary kiln process accounting for about 1% and smelting reduction contributing approximately 1%. The overarching current issues in the ironmaking industry are energy consumption and carbon dioxide emissions, which drive most of the new developments in ironmaking. National and international efforts in this respect include, among others, the American Iron and Steel Institute (AISI) CO$_2$ Breakthrough Program in the U.S., the ULCOS program in Europe, the COURSE 50 Program in Japan, the development of FINEX in Korea, and the efforts in Chinese steel industry [2]. As part of AISI's CO$_2$ Breakthrough Program, a novel Flash Ironmaking Technology (FIT) has recently been developed at the University of Utah. We will discuss this technology in some detail, vis-a-vis the current processes, from the viewpoint of CO$_2$ emissions and energy consumption.

### 1.1. The Blast Furnace (BF)

The modern blast furnace is a very large metallurgical reactor with a capacity of 0.50–5.6 million tons of pig iron per year. The largest of them is the No. 1 blast furnace at POSCO's Gwangyang (Korea) Steelworks that has the production capacity of 5.65 million tons per year [3].

In BF ironmaking, the solid feeds are fed from the top of a shaft furnace and preheated air is injected near its bottom. The solid charge consists of iron ore as sinter, pellets or lumps, mainly consisting of hematite ($Fe_2O_3$) or magnetite ($Fe_3O_4$), coke, and limestone as a flux. Preheated air is blown through tuyeres near the bottom to burn the coke, generating reducing CO gas and process heat. Coke maintains its strength at elevated temperatures, unlike coal that softens at the same temperatures. Iron ores must be sintered or pelletized to turn them into strong and porous pellets of 1–2 cm sizes to keep the bed permeable to the gas flow and facilitate the reduction.

Often, pulverized coal, natural gas, and/or oil are also injected together with the preheated air to decrease the consumption of expensive coke. The coke and injected fuels burn to produce combustion gas containing CO and $CO_2$ (>1800 °C). This hot gas flows up while heating the descending solids, causing partial reduction of iron oxides.

Liquid hot metal (pig iron), which typically contains 3.5–4.5 wt% of dissolved carbon, and molten slag collect in the bottom part (hearth) of the furnace.

### 1.2. Direct Reduction (DR)

In the Direct Reduction (DR) processes, iron ore is reduced to solid sponge iron with coal or a reducing gas mixture made up of $H_2$ and CO, most often produced from the reforming of natural gas, thus avoiding the use of coke. Worldwide, greater than 90% of the direct reduced iron (DRI) production is based on natural gas, whereas coal is mainly used in India [4].

The best-known technologies for DR are MIDREX [5] and HYL/Energiron [6], with MIDREX accounting for more than 78% of the DRI production in 2016. The shaft furnaces used in these technologies require pellets, which are 10–12 mm in size. Sponge iron produced from DR processes is fed to basic oxygen furnaces (BOF) and electric arc furnaces (EAF), as an alternative to BF because of its low capital cost and often based on the local conditions with respect to raw materials [4].

Other gas-based processes developed for the reduction of iron oxide but have been less adopted are the fluidized-bed processes FINMET, earlier FIOR [7], CIRCORED [8] and SPIREX [9]. These processes use iron ore fines, which are particles of +0.1 mm to −10 mm sizes, and thus provide low production rates because the particles in this size range react slowly. The process cannot be operated at high temperatures because the particles fuse and stick together at high temperatures. Largely because of these reasons, fluidized bed processes have not been very successful. The Flash Ironmaking Technology (FIT) utilizes iron ore concentrates with particle sizes less than 100 μm, which are smaller than fines by up to two orders of magnitude. These particles are reduced in seconds rather than the minutes and hours required in other processes.

The coal-based direct reduction process uses rotary kilns [10]. Although this process is slower and rather cumbersome, it is robust, uses less expensive non-coking coal instead of coke, operates at lower temperatures, and requires less feed preparation than most other ironmaking processes.

### 1.3. Smelting Reduction (SR)

Successful smelting reduction (SR) processes emerged during the 1990s. The main feature of SR is that pre-reduced iron ore is reduced by char generated from coal by in situ devolatilization to form molten metal and slag. This process bypasses cokemaking and requires less charge preparation, but usually needs a pre-reduction step. COREX, FINEX, and HISMELT are major examples. The more fully commercialized COREX and FINEX may be considered as processes that separate the shaft and the lower smelting sections of a blast furnace [11]. Pre-reduction of iron ore is first carried out, and then smelting is done in a separate melter-gasifier that contains a char bed continuously formed by

the devolatilization of coal continuously fed to the vessel. The COREX performs the pre-reduction in a shaft furnace, whereas the FINEX uses fluidized bed reactors in series for the same purpose. The melter-gasifier part is essentially the same in the two processes.

## 2. Critical Issues in Ironmaking

### 2.1. Technical Issues

World Steel Association [12] reports that the steel industry world-wide is responsible for 6.7% of the total $CO_2$ emissions. Further, the steel industry consumes the second largest amount of energy and emits the greatest volume of $CO_2$ (30%) of any industry. Lowering of energy use and $CO_2$ emissions in the conventional steelmaking processes is quickly approaching the theoretical bounds. To make significant further reductions in energy usage and $CO_2$ emissions, steelmaking will require drastically new ideas leading to the development of breakthrough technologies.

The BF operation has been greatly improved over the years with respect to efficiencies in productivity and energy use, coke rate, cokemaking technique, $CO_2$ emissions, and increased injection of other combustibles such as coal, natural gas, and plastics. Many of such technologies are already in commercial practice. However, the operation of a blast furnace still endures drawbacks related to high energy consumption, greenhouse gas emissions and large infrastructure cost.

The blast furnace is quite efficient from the viewpoint of energy and productivity, but requires pelletizing or sintering of iron ore and in addition must use coke as the main fuel and reducing agent. Cokemaking and sintering/pelletization are energy intensive and pollution prone. In addition, the use of coke generates large amounts of $CO_2$. Overall, sintering (13%), pelletization (2%), and cokemaking (5%) together produce ~20% of the overall $CO_2$ emissions in the BF-BOF route and the BF contributes ~70%, with 10% generated by steelmaking [13].

The utilization of iron ore fines or concentrates free of pelletizing and sintering is one of the alternate ways to lower $CO_2$ emissions and energy consumption [14]. These are important reasons for the development of and increasing attention paid to alternate ironmaking technologies. In addition to the above reasons, alternate processes like DR typically are more versatile and economical than BF at lower production rates. Despite these advantages, the current DR technologies based on shaft furnaces, rotary kilns, and fluidized bed reactors suffer from drawbacks such as a low energy efficiency when applied in a small scale, requirement for pelletization, and the fact that the produced DRI is pyrophoric and tends to re-oxidize. Furthermore, the rates of such processes are not intensive, for they cannot use higher temperatures because of the problems of solid sticking and fusion. The processes that use shaft furnaces, which dominate the DR industry, require the concentrate to be pelletized increasing operating costs and environmental problems. Currently, this process is less economical than the BF process as it requires iron ore of higher quality, limiting its flexibility [5].

Smelting reduction processes address many of the above-mentioned difficulties accompanying DR processes. However, they require more than one stage of operation and are unable to reduce $CO_2$ generation significantly because of their dependence on the use of coal.

### 2.2. Energy Requirements

Energy consumption is a critical issue in the steel industry. Table 1 shows a comparison between the four major steelmaking routes in terms of the energy consumed for the iron and steel production and also to generate electricity [15]. These numbers were based on the best practice of modern plants. In terms of just ironmaking, DR consumes the least amount of energy at 12.2 GJ/ton of steel and SR requires the largest amount at 18.7 GJ/ton, with BF positioned in the middle at 15.7 GJ/ton of steel. The use of an electric arc furnace for steelmaking is seen to be energy-intensive whereas the use of BOF for primary steelmaking requires a minimal amount of energy. In DR processes, iron oxide is reduced in solid state and therefore all gangues contained in the iron ore stay in the product iron (DRI) and

must be removed to the slag in the EAF. This causes greater electrical energy consumption than to melt scraps [4].

**Table 1.** Energy requirements by best practices worldwide for iron and steel production (GJ per tonne of mild steel) [15].

| Production Step | Process | BF-BOF | DR-EAF | SR-BOF * | EAF-Scrap |
|---|---|---|---|---|---|
| Feed preparation | Sintering | 2.2 | 2.2 | | |
| | Pelletizing | | 0.8 | 0.8 | |
| | Coking | 1.1 | | | |
| Ironmaking | | 12.4 | 9.2 | 17.9 | |
| Steelmaking | Main Step | −0.3 | 5.9 | −0.3 | 5.5 |
| | Refining | 0.4 | | 0.4 | |
| **Total (GJ/t)** | | **15.8** | **18.1** | **18.8** | **5.5** |

\* Mainly COREX; does not include FINEX or HISMELT.

**Note of Caution Regarding 'Energy Requirement':** It is noted here that a caution must be exercised when comparing the presented energy intensity values or energy requirements to clearly understand the energy items included in these values [16]. There are currently different approaches for selecting energy items to include in the overall "energy requirements" when an input fuel also serves as a reactant.

For a process involving heat generation from fuel combustion, the choice of which endothermic reactions to pick to compute the energy requirement can cause confusion. The essential question is whether the heating value of a reactant that is also burned to generate process heat should be added as an item in the 'energy requirement'. Some investigators include this item [17–19] and others do not [20–23]. The choice is arbitrary, but tone should indicate explicitly which approach is used in showing the energy calculations, especially in comparing different processes. The presented 'energy requirement' varies with the selected approach.

The net differences in the energy requirements between technologies are not strongly affected by the selection of calculation approaches if the same method is used for different technologies, but the consumption values for individual processes themselves depend on them. Thus, when presenting 'energy requirements', it is critical to definitively state the applied approach. Whether the heat of combustion of a reactant that can also be burned is included in the energy requirement should be clearly indicated. It will make it much more definitive to indicate the amounts for 'process energy' that contains only the heating value of the fuel and 'reductant energy' ('feedstock energy' in petrochemicals production) that includes the heating value of the substance serving as a reactant.

For detailed description of these different calculation methods and discussion on the subject, the reader is referred to Sohn and Olivas-Martinez [16]. An additional caution that follows from this consideration is that the comparison of different ironmaking processes for their energy requirements should only be done with the values obtained using the same method by the same investigators. In other words, the energy requirement value obtained by one investigator for a certain ironmaking process should not be compared with a value obtained by another investigator for a different ironmaking process without carefully checking the bases of the calculations.

Thus, the energy intensities listed in Table 1 for various technologies for making iron and steel may be used safely to obtain differences among the technologies. However, care should be exercised for example when using the energy intensity value for the BF-BOF route give in Table 1 to compare with the energy intensity of say the DR-EAF combination calculated or reported by a different investigator.

*2.3. Carbon Dioxide Emissions*

Greenhouse gas is a serious problem facing the world today. Although BF is a highly efficient reactor in terms of energy and chemical reactions, the coke used generates large $CO_2$ emissions. Steelmaking emits 1.9 tons of $CO_2$ for every ton of steel produced, accounting for 6.7% of the total man-made $CO_2$ emissions [12].

Although substantial decreases in emissions have been made, new steelmaking technologies will require drastically new ideas to make much greater reductions in energy consumption and $CO_2$ emissions.

As seen in Table 2, natural gas-based DRI production generates lower $CO_2$ emissions, in the range of 0.77–1.1 tons of $CO_2$ per ton of steel (in contrast to ~1.9 tons of $CO_2$ per ton of steel for the BF-BOF process), varying with the source of electricity used [1]. Smelting reduction (SR) processes like COREX and FINEX, which are coal-based, produce somewhat more $CO_2$ than DR processes but less than BF and much less than rotary kilns (SL/RN).

**Table 2.** $CO_2$ emissions from various steelmaking technologies in tons per ton of iron.

| BF-BOF [a] | Midrex Process [b] | HYL III-Energiron [a] | SL/RN Process [a] | SR [a,c] | Circored [a,d] |
|---|---|---|---|---|---|
| ~1.9 | ~1.1 | 0.77–0.92 | ~3.2 | 1.3–1.8 | ~1.2 |

[a] Institute for Industrial Productivity [4]; [b] Metius et al. [24]; [c] Hasanbeigi et al. [25]; [d] Husain et al. [8].

## 3. Development of Novel Flash Ironmaking Technology (FIT)

*3.1. Background*

Despite the many improvements and technical merits of the currently dominant blast furnace ironmaking process, an essential problem for the steel industry today is the development of a new technology with lower energy consumption, $CO_2$ emissions and fixed costs than the combined blast furnace and coke oven routes. An optimal process should also be able to produce at least 5000–10,000 tons of metal per day to be able to provide sufficient feed to the steel plants.

Given the limited potential for increased efficiency associated with current technologies and considering the other driving forces presented above, a Flash Ironmaking Technology (FIT) has recently been developed at the University of Utah as an innovative alternate ironmaking process [14,20–22,26–31]. This technology reduces iron ore concentrate in a flash reactor with a suitable reductant gas such as hydrogen or natural gas, and possibly bio/coal gas or a combination thereof. It is the first flash ironmaking process. This technology is suitable for an industrial operation that converts iron ore concentrate (less than 100 microns) to metal without further treatment. This transformative technology produces iron while bypassing pelletization or sintering as well as cokemaking steps that are energy intensive and pollution-prone. Further, the process is intensive due to the fact that the fine particles of the concentrate are reduced at a fast rate at 1200–1600 °C. Thus, the required residence times in this process is of the order of seconds rather than the minutes and hours required for pellets and even iron ore fines.

The fluidized bed processes, which have been less than successful commercially, also use fine iron ore particles. These ore 'fines' are several millimeters in size, compared with concentrate particles of <100 μm sizes. Thus, reduction takes much less time in FIT than in a fluidized bed reactor.

In the FIT concentrate particles are reduced by a hot gas mixture produced by the incomplete combustion of a fuel gas that serves as the source of heat as well as reductant gas ($H_2$ + CO + $H_2O$ + $CO_2$ mixture from natural gas and $H_2$ + $H_2O$ when pure hydrogen is used). A schematic of a possible large-scale flash ironmaking reactor is shown in Figure 1. The partial combustion of the feed gas produces the process heat and at the same time generates a gas mixture that has a sufficient reducing power for the reduction of iron oxide.

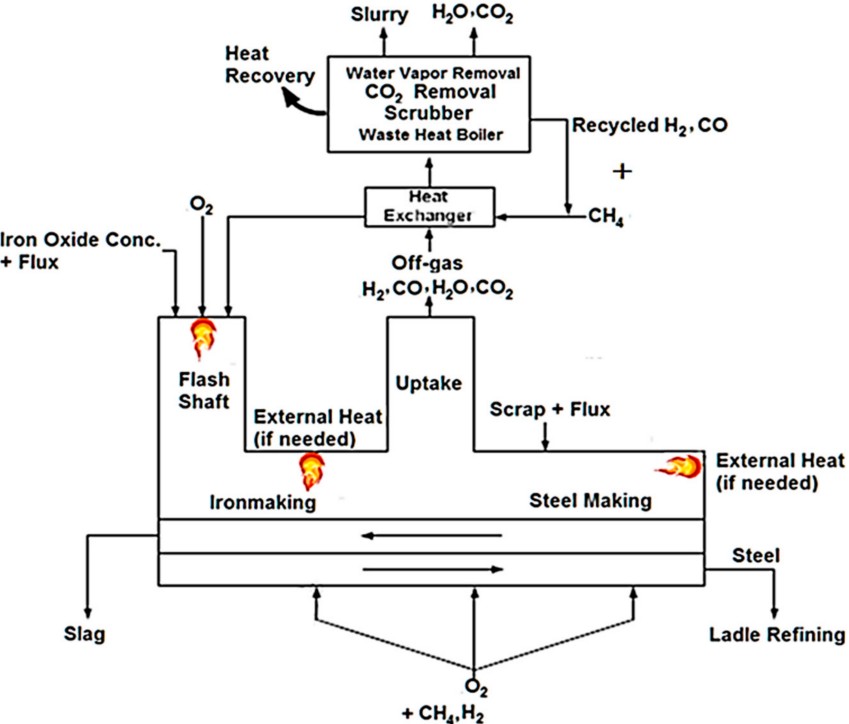

**Figure 1.** Flash Ironmaking Process. (Concept first proposed by H. Y. Sohn in a research proposal submitted to U.S. Department of Energy through American Iron and Steel Institute (AISI) in November 2003. The figure is adapted from Sohn and coworkers [28,32]).

The operating temperature of this process depends on the target final product. It can be operated at lower temperatures where iron is produced in solid state that can be fed to downstream steelmaking processes such as EAF. In this case, the flash reactor would mainly consist of the shaft portion of the facility shown in Figure 1. In a high-temperature operation, where iron is produced in a molten state, the process can be used as an overall continuous direct steelmaking process, as depicted in Figure 1 [28,32].

The use of concentrate particles and natural gas eliminates the pelletization and cokemaking steps, which lowers the energy consumption by 30% [16] and $CO_2$ emissions by 39–51% relative to the average blast furnace process [20–22]. If the technology is operated with hydrogen produced from sources with no carbon footprint, the $CO_2$ emissions will be lowered to 4% (from limestone calcination) of that from the blast furnace process.

In the U.S., iron ore is mostly produced from the Mesabi Range in Minnesota and Michigan, which made up 97% of the iron ore products in the U.S. in 2017 [33]. The ore is largely taconite mineral, a low-grade magnetite ore containing 15–30 wt% iron that requires upgrading for industrial use by crushing and grinding to liberate the iron-bearing minerals and concentrating to remove the gangue materials (mainly silica) by magnetic separation and flotation. The concentrate particles are less than 100 μm in size, whereas the feed to the blast furnace must be coarser than 2 mm. Therefore, the concentrate must be pelletized in the BF process by first forming green pellets [34] and then hardening them by an induration process in which they are dried and fired at ~1300 °C.

### 3.2. Energy Requirements

Tables 3 and 4 list energy input and output items, respectively, for the flash ironmaking process using in situ partial combustion of natural gas or hydrogen, compared with those for the average blast furnace operation. The energy values in these tables include the heat of combustion of fuel/reductant consumed for the reduction of iron oxide plus that used for process heating. The numbers in these

tables are for the production of 1 million tons of iron per year. The values for the flash ironmaking process are based on the flow sheets developed by Pinegar et al. [20,21]. The values for an average blast furnace operation were calculated using published data and applying the same method as for the flash ironmaking process. Sohn and Olivas-Martinez [16] also presented the energy balances for these processes as Sankey diagrams.

**Table 3.** Energy input for an industrial-scale flash ironmaking process using natural gas or hydrogen vs. an average blast furnace process (for production of 1 million tons of iron). (Adapted from Pinegar et al. [20,21]).

| | Process | Reformerless Natural Gas | Hydrogen [d] | Blast Furnace [a,e] |
|---|---|---|---|---|
| | Fuel combustion [b] | 19.22 | 14.05 | 13.60 |
| | Heat recovery (sum of next 2) | −4.77 | −2.80 | −1.32 |
| | (Waste heat boiler) | (−3.39) | | |
| ITEMIZED INPUT | (Steam not used) | (−1.38) | | |
| (GJ/ton Fe) | Sub-total | 14.45 | 11.25 | 12.28 |
| | Ore/Coke preparation [c] | | | 5.68 |
| | $CaCO_3$ and $MgCO_3$ calcination (external) | 0.26 | 0.26 | |
| | Total | 14.7 | 11.5 | 18.0 |

[a] Energy balance was calculated by METSIM based on the published material balance. [b] Higher heating values of the natural gas and hydrogen were used for this calculation. [c] See Sohn and Olivas-Martinez [16] for references. [d] Energy requirement for hydrogen production was not included for this calculation. The energy requirement for hydrogen production depends on the production process such as steam-methane reforming, coal gasification or water splitting. [e] In fairness to excluding the energy requirement of hydrogen production for flash ironmaking, the energy requirement for producing coking coal was not included for the blast furnace.

**Table 4.** Energy output for an industrial-scale flash ironmaking process using natural gas or hydrogen vs. an average blast furnace process (for production of 1 million tons of iron). (Adapted from Pinegar et al. [20,21]).

| | Process | Reformerless Natural Gas | Hydrogen | Blast Furnace [a] |
|---|---|---|---|---|
| | Reduction [b] | 6.68 | 6.68 | 7.37 |
| | Sensible heat of iron | 1.27 (1773 K) | | 1.35 (1873 K) |
| | Sensible heat of slag | 0.24 (1773 K) | | 0.47 (1873 K) |
| | Slurry ($H_2O$ (l)) | 2.25 (323 K) | 1.93 | |
| | Hot water not used | 1.57 (493 K) | | |
| | Flue gas | 0.79 (573 K) | | 0.26 (363 K) |
| | Removed water vapor | | 0.01 | |
| | $CaCO_3$ decomposition | | | 0.33 |
| | Slagmaking | | | −0.17 |
| ITEMIZED OUTPUT | Heat loss in the reactor | 0.78 | 0.78 | 2.60 |
| (GJ/ton Fe) | Heat loss in the heat exchangers (sum of next 3) | 0.73 | 0.34 | 0.07 |
| | (Reactor feed gas heater) | (0.40) | | |
| | (Natural gas heater) | (0.21) | | |
| | (WGS reactor feed gas heater) | (0.12) | | |
| | Steam not used (363 K) | 0.14 | | |
| | Sub-total | 14.45 | 11.25 | 12.28 |
| | Pelletizing [c] | | | 3.01 |
| | Sintering [c] | | | 0.65 |
| | Cokemaking [c] | | | 2.02 |
| | $CaCO_3$ and $MgCO_3$ calcination (external) | 0.26 | 0.26 | |
| | Total | 14.7 | 11.5 | 18.0 |

[a] Energy balance was calculated by METSIM based on the published material balance. [b] For flash ironmaking process, magnetite was used as the iron oxide; hematite was used for the blast furnace, because magnetite in the concentrate is converted to hematite during pelletization required in the blast furnace (BF) process. [c] See Sohn and Olivas-Martinez [16] for references.

In the blast furnace process, the energy needed for sintering, pelletizing and cokemaking operations, which are not necessary in flash ironmaking, represents a large part of the overall energy requirement. The preparation of ore and coke takes up approximately 30% of the total energy input [16].

The 'energy requirement' for ironmaking can be calculated directly from the sums of energy inputs in Table 3 and outputs in Table 4. This is because the recovered heat is listed as a negative input item in Table 3. The energy balance could be written with such an item listed as a positive output term in Table 4, in which case the 'balance' is still attained.

The total value in these tables for BF is somewhat different from that for the ironmaking portion given in Table 1. As noted in that section, energy requirement values depend on the items included in the calculation procedure. However, the differences in the values computed by applying the same calculation procedure remain the same. According to Table 1 a typical current Direct Reduction process for ironmaking consumes approximately 3.5 GJ/t Fe less than BF [15], which would correspond to 14.5 GJ/t Fe if calculated using the same basis for obtaining the results in Tables 3 and 4. This number agrees with the energy requirement for the reformerless mode of the Flash Ironmaking Technology.

### 3.3. Carbon Dioxide Emissions

When natural gas is the fuel/reductant in the Flash Ironmaking Technology (FIT), the level of carbon dioxide emissions is similar at ~1 ton/ton Fe [21] to those from other natural gas based DR processes listed in Table 2. However, a hydrogen-based flash ironmaking process would emit a lower amount of $CO_2$, depending on the source of hydrogen.

### 3.4. Reduction Kinetics of Concentrate Particles

A flash reactor typically provides a residence time measured in seconds, unlike a shaft or fluidized bed reactor that can provide a residence time of minutes and hours. Thus, an essential requirement in the development of a flash ironmaking process is sufficiently fast reduction rate of the iron oxide feed. In a fluidized bed reactor, the iron ore fines require residence times measured in minutes. It is important, however, to differentiate concentrates from iron ore fines. The majority of individual particles in fines are several millimeters in size, while concentrate particles are less than 100 μm. There have been many applications of iron ore fines in ironmaking such as FIOR, FINMET, Circored, Iron Carbide and the recent FINEX processes. On the other hand, there have not been any processes directly using iron ore concentrates in large scale.

Previous researchers have considered the in-flight reduction of iron oxide particles, but the reduction rate was deemed too slow for a flash process that typically provides a residence time of a few seconds. This conclusion was based on data from the reduction of particles ranging from 70 to 42,000 μm at temperatures 600–1000 °C [35]. Sohn [14] examined these data and concluded that there was a potential for concentrate particles to be reduced in a few seconds. He then launched a project to develop the novel Flash Ironmaking Technology (FIT). It was thus important to establish a sufficiently fast rate for the reduction of iron concentrates using $H_2$ and CO gas mixture for designing the flash reactor.

Therefore, the reduction kinetics were investigated in the temperature range 1150–1600 °C using $H_2$, CO, or $H_2$ + CO as the reducing gas by Sohn and coworkers [36–42]. Reduction by $H_2$ + CO is complicated because of the simultaneous reduction by the two gases as well as the water-gas shift reaction involving CO and $H_2$ with the product gases $H_2O$ and $CO_2$.

Due to the melting of the particles above 1350 °C, the rate analysis was done separately in the ranges of 1423–1623 K (1150–1350 °C) and 1623–1873 K (1350–1600 °C). It is seen in Figure 2 that the particles maintain their original solid shape in the lower temperature range, whereas they melt into spheres at higher temperatures, as seen in Figure 3.

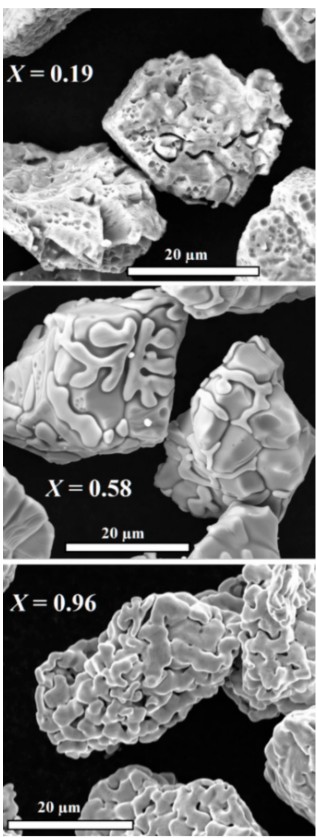

**Figure 2.** SEM micrographs of particles at 1137–1324 °C. (*X* is fractional conversion.) (The figure is adapted from [39], with permission from John Wiley & Sons, 2017).

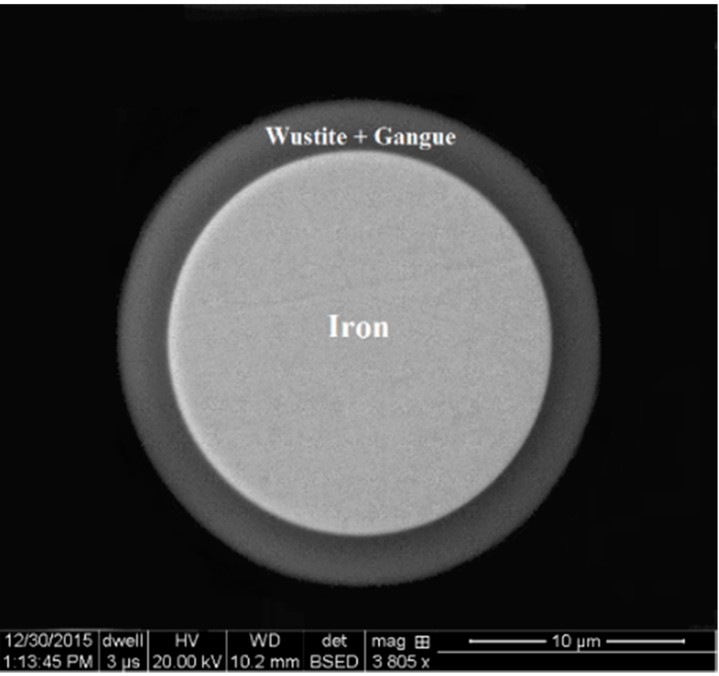

**Figure 3.** Cross section of a particle reduced to fractional conversion *X* = 0.94 at 1623 K (1350 °C) (The figure is adapted from [43], with permission from authors, 2019).

At these temperatures and particle sizes, it was proved that neither external mass transfer nor pore-diffusion influence the rate of magnetite concentrate reduction, as the reaction would be

completed in milliseconds if the rate was diffusion-controlled compared with several seconds observed experimentally. Therefore, the rate equations obtained based on the experimental data are those of the chemical reactions unaffected by mass transfer [28]. The reduction of magnetite to iron proceeds through the formation of $Fe_{0.947}O$ in the temperature range of flash smelting. However, it is not possible to measure the separate kinetics of $Fe_{0.947}O$ and Fe formation for fine particles reacting rapidly. Moreover, different parts of a fine irregular particle react at different rates and thus various oxide phases may coexist at any time, as the SEM and XRD results of quenched samples in Figure 4 show. Thus, a global rate equation was used in which conversion represented the fraction of oxygen removed from the original iron oxide without recognizing individual oxide phases.

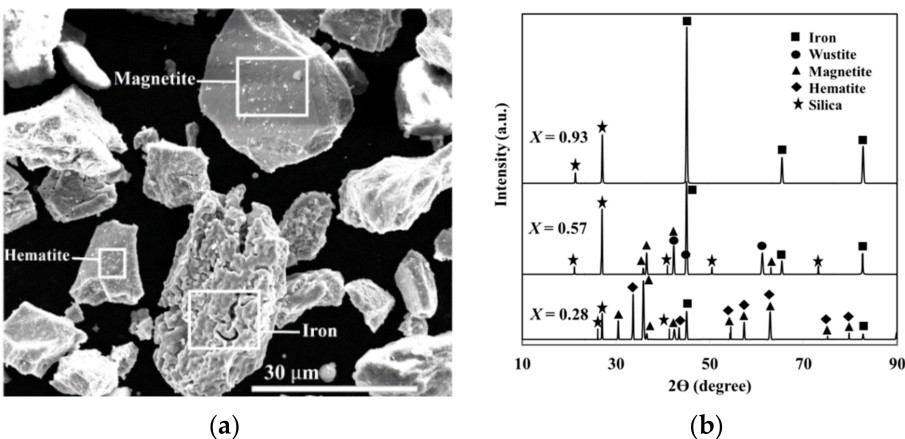

(**a**)　　　　　　　　　　　　　　　　　　　　　　　(**b**)

**Figure 4.** Existence of different iron oxide phases indicated by: (**a**) SEM micrographs of reduced particles (X = 0.14) and (**b**) XRD of samples at various reduction degrees. (The figure is adapted from [40], with permission from Springer Nature, 2015).

The resulting rate expressions from all the work mentioned above are summarized below:
***Reduction of Magnetite Concentrate by H₂***
In temperature range 1423–1623 K (1150–1350 °C) [38]:

$$\left.\frac{dX}{dt}\right|_{H_2} = 1.23 \times 10^7 \exp\left(-\frac{196,000}{RT}\right) \cdot \left(p_{H_2} - \frac{p_{H_2O}}{K_{H_2}}\right) \cdot (1-X) \tag{1}$$

where $t$ is in seconds, $R$ is 8.314 J/mol K, $T$ is in K, and $p$ is in atm.
In temperature range 1623–1873 K (1350–1600 °C) [43]:

$$\frac{dX}{dt} = 6.07 \times 10^7 \cdot e^{-\frac{180,000}{RT}} \cdot \left(p_{H_2} - \frac{p_{H_2O}}{K}\right) \cdot \left(d_p\right)^{-1} \cdot (1-X) \tag{2}$$

where $d_p$ is in μm.
***Reduction of Magnetite Concentrate by CO***
In temperature range 1473–1623 K (1200–1350 °C) [44]:

$$\frac{dX}{dt} = 5.35 \times 10^{13} \cdot e^{-\frac{451,000}{RT}} \cdot \left(p_{CO} - \frac{p_{CO_2}}{K}\right) \cdot (1-X) \cdot [-\ln(1-X)]^{-1} \tag{3}$$

In temperature range 1623–1873 K (131,600 °C) [44]:

$$\frac{dX}{dt} = 3.225 \times 10^3 \cdot e^{-\frac{88,000}{RT}} \cdot \left(p_{CO} - \frac{p_{CO_2}}{K}\right) \cdot \left(d_p\right)^{-1} \cdot (1-X) \cdot [-\ln(1-X)]^{-1} \tag{4}$$

*Reduction of Magnetite Concentrate by H₂ + CO Mixture* [45]

$$\frac{dX}{dt} = \left(1 + 1.3 \cdot \frac{p_{co}}{p_{co} + p_{H_2}}\right) \cdot \left.\frac{dX}{dt}\right|_{H_2} + \left.\frac{dX}{dt}\right|_{CO} \qquad 1423\text{ K} < T < 1623\text{ K} \tag{5}$$

$$\frac{dX}{dt} = \left[1 + (-0.01T + 19.65) \cdot \frac{p_{co}}{p_{co} + p_{H_2}}\right] \cdot \left.\frac{dX}{dt}\right|_{H_2} + \left.\frac{dX}{dt}\right|_{CO} \qquad 1623\text{ K} < T < 1873\text{ K} \tag{6}$$

where $\left.\frac{dX}{dt}\right|_{H_2}$ and $\left.\frac{dX}{dt}\right|_{CO}$ are the instantaneous rates of reaction obtained from the corresponding reaction rates individually by $H_2$ and by CO, respectively, given in Equations (1)–(4).

*Reduction of Hematite Concentrate by H₂* [36]

$$\frac{dX}{dt} = 8.47 \times 10^7 \times e^{-\frac{218,000}{RT}} \cdot \left[p_{H_2} - \left(\frac{p_{H_2O}}{K_{H_2}}\right)\right] \cdot (1 - X) \qquad 1423\text{ K} < T < 1623\text{ K} \tag{7}$$

*Reduction of Hematite Concentrate by CO* [36]

$$\frac{dX}{dt} = 5.18 \times 10^7 \times e^{-\frac{241,000}{RT}} \cdot \left[p_{CO} - \left(\frac{p_{CO_2}}{K_{CO}}\right)\right] \cdot (1 - X) \qquad 1423\text{ K} < T < 1623\text{ K} \tag{8}$$

*Reduction of Hematite Concentrate by H₂+CO Mixture* [37]

$$\frac{dX}{dt} = \left[1 + (-0.004T + 7.004) \cdot \frac{p_{co}}{p_{co} + p_{H_2}}\right] \cdot \left.\frac{dX}{dt}\right|_{H_2} + \left.\frac{dX}{dt}\right|_{CO} \qquad 1423\text{ K} < T < 1623\text{ K} \tag{9}$$

with $\left.\frac{dX}{dt}\right|_{H_2}$ and $\left.\frac{dX}{dt}\right|_{CO}$ given by the instantaneous rates of reaction obtained from the reaction rates individually by $H_2$ and by CO, respectively, given in Equations (7) and (8).

A serious problem with the direct reduced iron is its pyrophoric nature due to the morphology and large specific surface area when produced below about 900 °C. Sohn and co-workers [46,47] investigated the re-oxidation rates of the iron produced in the flash ironmaking process under various oxidizing gases. The re-oxidation rate by $H_2O$ was determined under various temperatures and water partial pressures at 550–700 °C and $H_2O$ concentrations of 40–100%. During the few seconds available in a flash process, the re-oxidation extent of iron particles in water vapor was <0.24%. The investigation of the oxidation of iron particles by an $O_2$-$N_2$ gas atmosphere was also performed and it was concluded that the flash-reduced iron was not pyrophoric, unlike the direct reduced iron. The flash ironmaking takes place at higher temperatures than the direct reduction and thus the surface of the flash reduced iron was passivated, as seen from the micrographs in Figure 5. This figure shows iron particles reduced at a lower temperature, like conventional DRI (Figure 5a), compared with those obtained at a flash ironmaking temperature (Figure 5b). This work established that the oxidation of flash reduced iron does not pose any concern at temperatures below 573 K (300 °C) [46].

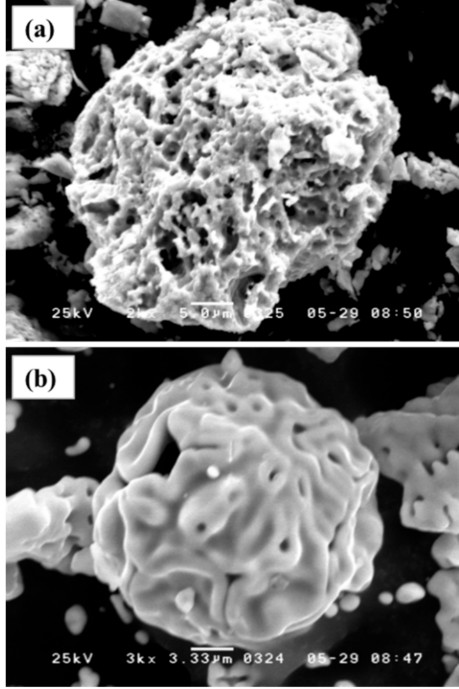

**Figure 5.** Microstructures of iron particles: (**a**) reduced by hydrogen at 1073 K (800 °C); (**b**) flash reduced at 1623 K (1350 °C). (The figure is adapted from [46], with permission from authors, 2014).

### 3.5. Laboratory Flash Reactor

An industrial flash reactor would be quite different from the laminar-flow reactor used for the rate measurement, including the fact that an oxy-fuel burner would be the main source of heat and the amount of excess reducing gases would be much lower (20–100%). Therefore, a laboratory flash reactor was installed at the University of Utah [29,31,48] to test the Flash Ironmaking Technology (FIT). The experiments performed in this reactor aimed to establish the optimum conditions for the design of the industrial flash reactor. In this reactor, magnetite concentrate was reduced by a reducing gas mixture generated from the partial oxidation of methane and/or hydrogen with industrial oxygen, which provided heat and produced a reducing gas mixture of $H_2$ and CO.

The apparatus, shown in Figure 6, consisted of an electrical furnace housing a stainless-steel tube, a gas delivery system, a powder feeding system, a power control system, an off-gas scrubbing system, and an off-gas burner. The electrical furnace housed a 316 stainless-steel tube with 19.5 cm ID and 213 cm length.

The particles were fed into this reactor through openings in the upper flange installed on the top of the reactor tube. Figure 7 shows the two feeding modes tested in this work: (a) feeding through the center of the fuel/oxygen burner; (b) feeding through two ports on opposite sides of the burner.

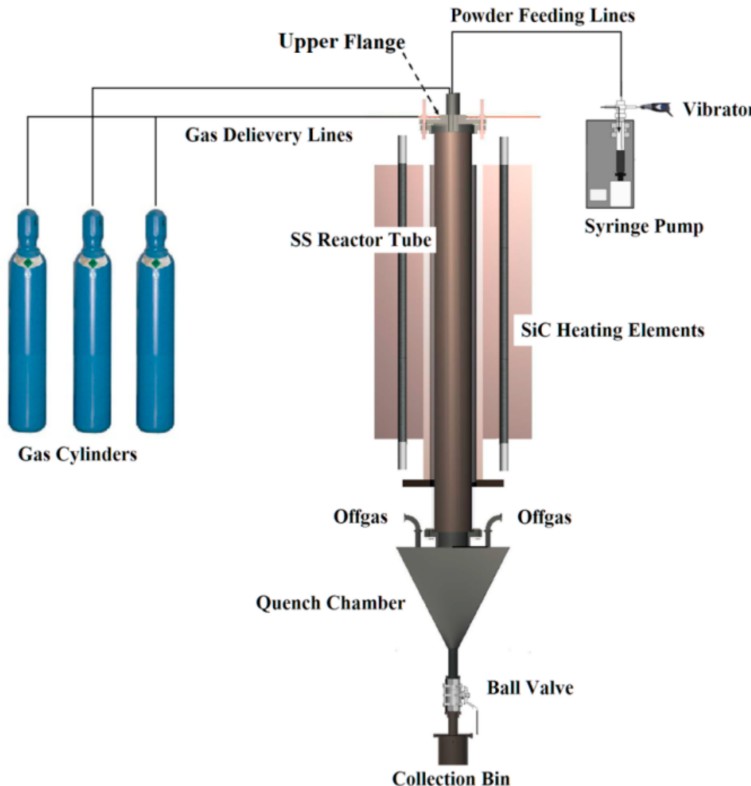

**Figure 6.** Schematic diagram of the laboratory flash reactor. (The figure is adapted from Sohn et al. [31]).

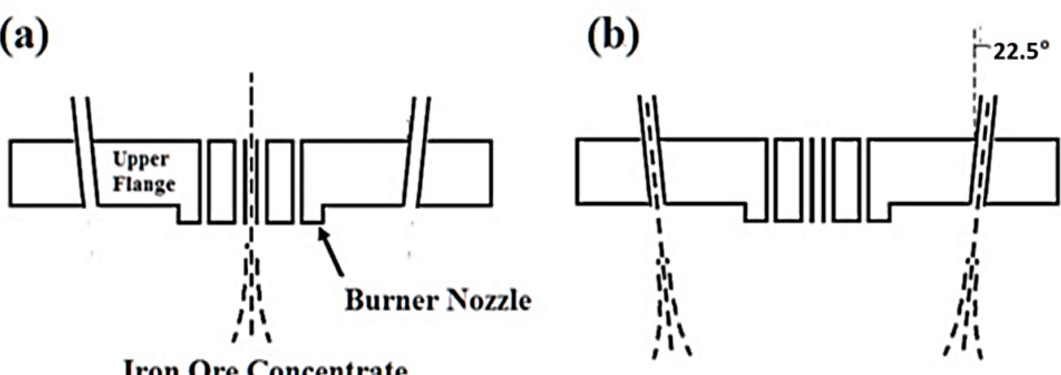

**Figure 7.** Powder feeding modes: (**a**) Burner feeding port. (**b**) Two Side-Feeding ports. (The figure is adapted from Sohn et al. [31]).

The burner was made of Inconel with crescent-shaped feeding inlets (Slots 1) and cylindrical inlets (Slots 2), as shown in Figure 8. Two different flame configurations were tested by switching the fuel and oxygen injection slots. F-O-F (F = fuel; O = oxygen) was the first configuration where the fuel (hydrogen or methane) was injected through Slots 1 and surrounded the oxygen injected through Slots 2. O-F-O was the second configuration where the oxygen was injected from Slots 1 surrounding the fuel injected from Slots 2.

The variation in the burner configuration changed the temperature distribution in the upper part of the reactor. Computational fluid dynamics (CFD) simulations [48,49] indicated that variation in the flame configuration affected the best feeding modes as the particles experience different temperatures in the different flame configurations, as will be discussed subsequently.

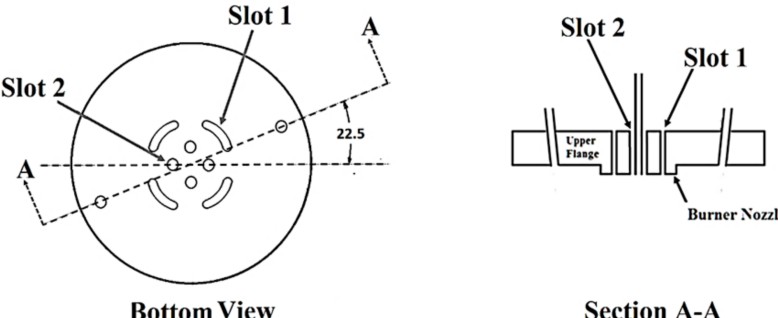

**Figure 8.** Schematic diagram showing the fuel/oxygen burner: **Left figure**-Bottom view. **Right figure**-Cross section at A-A. (The figure is adapted from Sohn et al. [31]).

### 3.5.1. Experiments with Hydrogen

A reduction degree of >90% with <100% excess hydrogen at temperature as low as 1175 °C was achieved in a few seconds of residence time. Figure 9 shows the effect of residence time and excess driving force (EDF) on the extent of reduction. EDF, which represents the level of excess hydrogen fed, is defined as follows [28]:

$$
\text{EDF} = \frac{\left(\frac{p_{H_2}}{p_{H_2O}}\right)_{\text{actual}} - \left(\frac{p_{H_2}}{p_{H_2O}}\right)_{\text{equ.}}}{\left(\frac{p_{H_2}}{p_{H_2O}}\right)_{\text{equ.}}} = K_H\left(\frac{p_{H_2}}{p_{H_2O}}\right)_{\text{actual}} - 1 \tag{10}
$$

where $p_{H2,\text{actual}}$ and $p_{H2O,\text{actual}}$ are the partial pressures of $H_2$ and $H_2O$ in the gas mixture at complete reduction, respectively, and $p_{H2,\text{equ.}}$ and $p_{H2O,\text{equ.}}$ are the partial pressures of $H_2$ and $H_2O$ at equilibrium with wüstite and Fe.

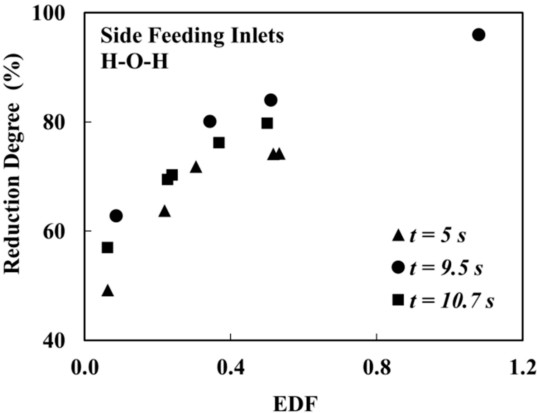

**Figure 9.** Effect of excess driving force (EDF) on reduction degree (%) when feeding through the two side-ports with the H-O-H configuration [31].

The H-O-H flame generated a higher local temperature in the middle of the flame compared to the O-H-O flame [31]. When the particles were injected through the burner in the H-O-H flame arrangement, they melted and then solidified into spherical particles. Melting reduced the surface area in contrast to side feeding where the particles retained their irregular shape and reactivity that resulted in higher reduction degrees. Therefore, the reduction degree obtained from the burner feeding was lower than that obtained from the side feeding at the same experiment conditions.

The temperature in the middle of the flame with the O-H-O arrangement was lower than with the H-O-H arrangement, 1150 °C and 2577 °C, respectively. Therefore, particle melting did not occur

and they kept their irregular shape even when going through the flame, which resulted in a higher reduction degree. Figure 10 illustrates the change in the particles shape with the flame arrangement at otherwise the same conditions.

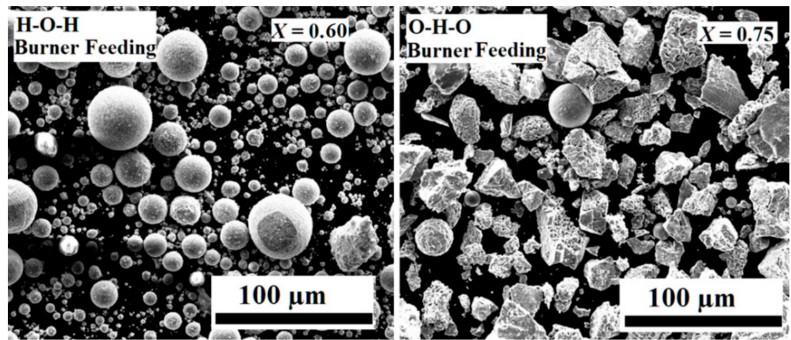

**Figure 10.** SEM micrographs of the particles collected from experiments with feeding through the burner using H-O-H (Reduction degree = 60%) (**Left**) and O-H-O (Reduction degree = 75%) (**Right**) configurations. (The figure is adapted from Sohn et al. [31]).

### 3.5.2. Experiments with Methane

Experiments were performed in the laboratory flash reactor in which natural gas was partially burned with industrial oxygen, producing heat and $H_2 + CO$. It was determined that at above 1150 °C, the reduction in an $H_2 + CO$ gas mixture is done mainly by hydrogen.

To set the experimental conditions, the HSC 5.11 thermodynamics software was used to calculate the equilibrium product composition. Using this calculated equilibrium composition, the nominal residence time of the particles and the excess driving force (EDF) with respect to $H_2$ were calculated.

The O-M-O configuration was used in the case of solid feeding through the center of the burner to avoid the melting of the concentrate particles. The O-M-O configuration yielded a higher reduction degree compared with the M-O-M configuration, as shown in Figure 11. When feeding through the side ports, the particles experienced higher temperatures in the O-M-O flame compared to the M-O-M flame [31].

The results obtained from this reactor proved that iron oxide could be reduced directly in a flash reactor utilizing natural gas or hydrogen. These results were used to determine the best feeding modes and the flame configuration to be implemented in a larger scale reactor as well as an industrial reactor. Magnetite reduction degree >90% was obtained at a temperature as low as 1175 °C with 100% excess hydrogen driving force or less in a few seconds of residence time.

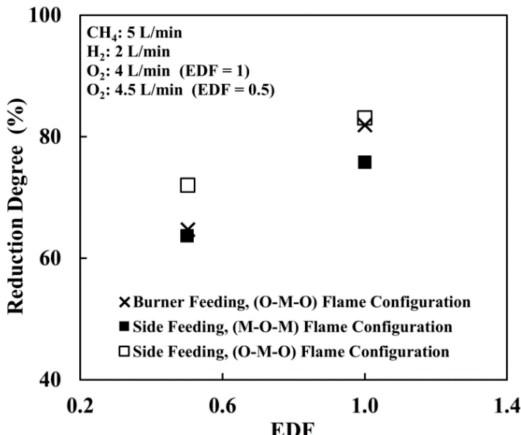

**Figure 11.** Effect of EDF on reduction with different feeding modes and flame arrangements. (The figure is adapted from Sohn et al. [31]).

### 3.6. Mini-Pilot Reactor Testing

In this section, we describe a mini-pilot reactor called the Large-Scale Bench Reactor (LSBR) operating at 1200–1600 °C and a concentrate feeding rate of 1–7 kg/h, installed at the University of Utah and shown in Figure 12.

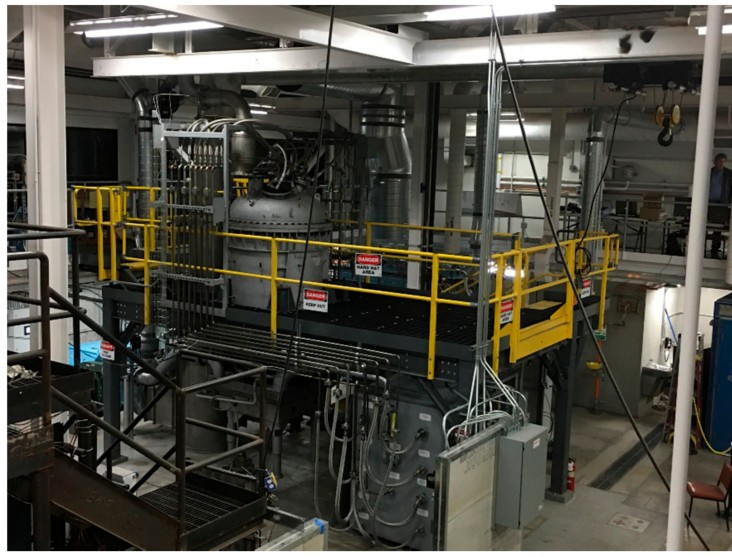

**Figure 12.** The Large Bench Reactor installed in the University of Utah.

The LSBR consists of a reactor vessel, a vessel roof with various feeding and auxiliary ports, burners, a quench tank, off-gas piping, a flare stack, an off-gas analyzer, a gas valve train, a water cooling system, gas leak detectors, a concentrate feeding system, and human machine interface. Figure 13 shows the main components of the reactor body.

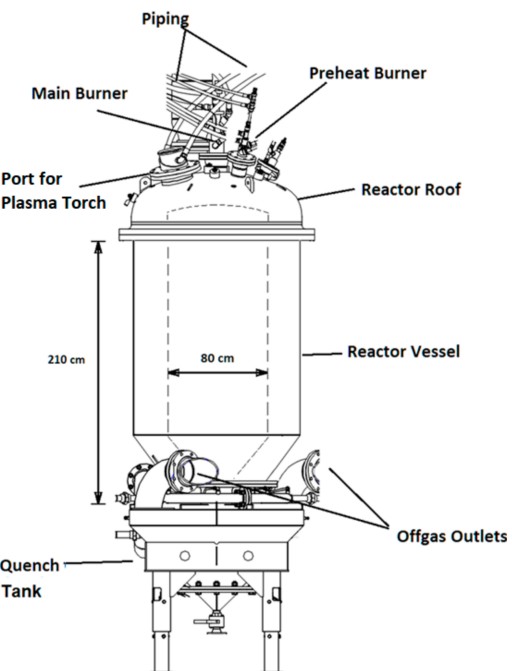

**Figure 13.** Schematic diagram of the Large-Scale Bench Reactor (LSBR).

The reactor vessel was built with a carbon steel shell and lined with three wall layers: 0.3 cm of alumina-silica fiber blanket, 8 cm layer of high fired and pressed silica, and 18 cm of 99.8% alumina castable refractory layer with high hot strength. The inner diameter of the vessel was 80 cm and the length was 210 cm, as shown in Figures 13 and 14. Figure 14 also shows photos of the main reactor and the burner as well as other details by schematics.

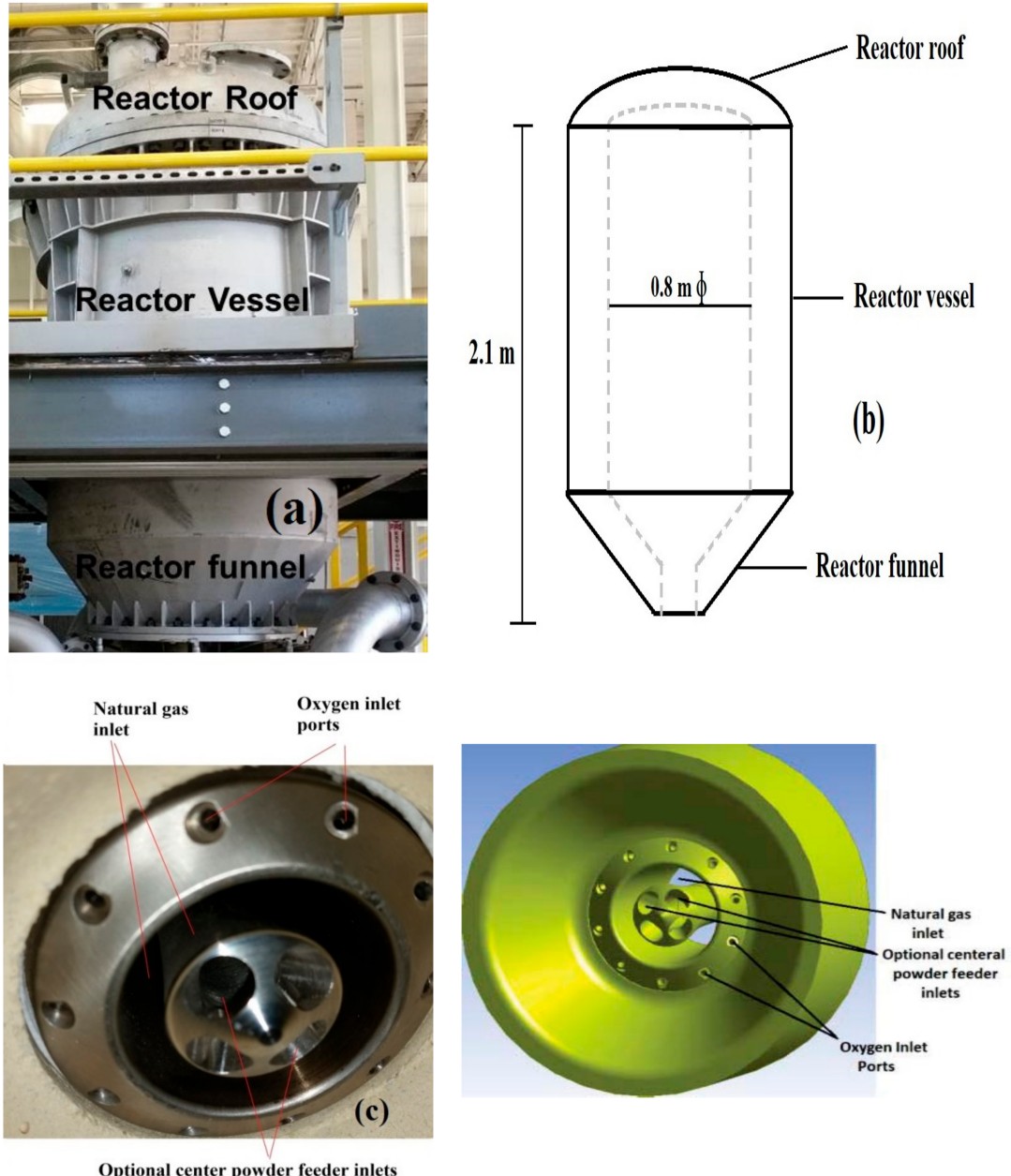

**Figure 14.** (**a**) The Large-Scale Bench Reactor (LSBR) Vessel, (**b**) Schematic diagram of the LSBR, (**c**) The burner for the LSBR, and Schematic of the burner (**Lower right**).

The LSBR had three burners: a preheat burner, the main burner, and a plasma torch. The preheat burner was used for preheating the reactor to the target temperature using a natural gas and industrial oxygen combustion flame. The preheat burner contained a pilot burner that generated a small flame to ignite the preheat burner. The pilot burner had a fiberglass flame detector that detected the pilot flame and started the flow of natural gas and oxygen through the preheat burner. Through the main burner, natural gas and industrial oxygen were injected to produce a flame and reducing gas mixture by partial

oxidation. The plasma torch was installed to provide heat, if needed, without affecting the process gas flow rates.

The Human Machine Interface consisted of the programmable logic controller (PLC) and a computer. The main PLC was connected to all the different parts of the system and to the computer where the operator could monitor the various parts and run the reactor. The programming in the main PLC was responsible for all the safety and emergency steps.

During preheating, combustion of natural gas with industrial oxygen was used to heat the reactor at a ramp rate of 90–95 °C/h to the operating temperature of 1200–1550 °C. The reactor was preheated by burning natural gas with oxygen. The wall temperature was measured by B-type thermocouples imbedded 2.5 cm inside of the inner wall surface. The concentrate was fed to the reactor using a pneumatic feeder with a rate of 1–7 kg/h through 2 feeding ports 0.28 m away from the center of the roof.

The gas analyzer used an NDIR (infrared) detector for measuring the $CO$, $CO_2$ and $CH_4$ contents, a thermal conductivity cell for measuring the $H_2$ concentration, and an electrochemical sensor for measuring the $O_2$ content. The partial oxidation of natural gas produced $H_2$ and $CO$, but $CH_4$ and $O_2$ contents in the off-gas were always less than the detection limits.

The experimental conditions were selected to represent the industrial conditions and to produce a gas mixture that had sufficiently high reducing power and temperature. In addition, the gas flow rates were set to generate sufficient heat and residence time for high degrees of reduction.

The results of the LSBR runs will help in designing an industrial reactor in terms of the identification of the technical hurdles and improvement of the operation. This reactor was simulated by 3-D CFD to optimize the operating conditions for an industrial reactor [50]. Six runs with the LSBR listed in Table 5 were simulated. The results obtained from the CFD model was in satisfactory agreement with the results of the reactor runs, especially considering the complexity of the process and the size of the facility. This work further identified potential safety issues and solutions that are needed in the design and operation of an industrial flash ironmaking reactor.

**Table 5.** Experimental conditions of the LSBR.

| Parameters | Run 1 | Run 2 | Run 3 | Run 4 | Run 5 | Run 6 |
|---|---|---|---|---|---|---|
| Concentrate feeding rate, kg h$^{-1}$ | 2.5 | 4.3 | 5.0 | 5.0 | 4.6 | 4.0 |
| Particle size range, μm | 32–90 | less than 90 | less than 90 | 32–90 | less than 90 | less than 90 |
| Mass average particle size, μm(used for simulation) | 45 | 32 | 32 | 45 | 32 | 32 |
| Natural Gas flow rate, [a] m$^3$ h$^{-1}$ | 25.16 | 30.56 | 20.36 | 24.80 | 17.36 | 15.86 |
| Natural Gas input temperature, K | | | | 300 | | |
| $O_2$ flow rate, [a] m$^3$ h$^{-1}$ | 19.85 | 19.67 | 14.27 | 21.53 | 16.35 | 14.81 |
| $O_2$ input temperature, K | | | | 300 | | |
| Total inlet gas flow rate, [a] m$^3$ h$^{-1}$ | 45.01 | 50.23 | 34.63 | 46.33 | 33.71 | 30.67 |
| $O_2$ to Natural Gas mole ratio | 0.79 | 0.64 | 0.70 | 0.87 | 0.94 | 0.93 |
| Inner wall temperature, [b] K | 1483–1563 | 1503–1603 | 1573–1673 | 1403–1473 | 1563–1623 | 1573–1623 |
| Inner wall temperature, K (used for simulation) | 1526 | 1548 | 1626 | 1440 | 1594 | 1599 |

[a] Volumetric flow rates are presented at 298 K and 0.85 atm, the atmospheric pressure at Salt Lake City (1 atm = 101.32 kPa). [b] The measured wall temperature varied in the shown range during a run and the average value was selected for Computational fluid dynamics (CFD) simulation.

### 3.7. Computational Fluid Dynamics Simulation

3-D Computational fluid dynamics (CFD) technique was used to simulate the fluid flow, heat transfer and chemical reaction of the concentrate in the shaft of a flash ironmaking reactor. Flash reactor runs to test the effects of different powder feeding schemes, different flame configurations, and hydrodynamic conditions of an industrial flash ironmaking reactor were simulated using the CFD technique. Temperature and species contours, gas flow patterns, and particle trajectories inside the reactor were computed while incorporating the rate expressions.

The Euler-Lagrange approach was used in this simulation, in which the gas phase was described in the Eulerian frame of reference while the particles were tracked in the Lagrangian framework. Particle spread by turbulence was described using the stochastic trajectory model. Detailed description of the CFD model for LSBR can be found elsewhere [50].

Experimental reduction degrees and the corresponding computed values are presented in Table 6. A reasonable agreement in the reduction degrees is seen in the first three runs. The disagreements for the last three runs were attributed to possible particle agglomeration at higher temperatures, which was not considered in the simulation [50].

**Table 6.** Experimental vs. computed reduction degrees.

| Run | Experimental (pct) | Simulation (pct) |
|---|---|---|
| 1 | 94.0 | 99.8 |
| 2 | 80.0 | 84.5 |
| 3 | 94.5 | 99.6 |
| 4 | 74.0 | 99.8 |
| 5 | 72.5 | 99.5 |
| 6 | 50.0 | 85.0 |

### 3.8. Economic Analysis

Sohn and coworkers [20–22,51] studied the economic and environmental aspects of the FIT by using the METSIM software to assess different process configurations for a plant that produces 1 million tons per year of solid iron powder. These authors constructed the flow sheet for an industrial-scale plant based on the Flash Ironmaking Technology and carried out detailed material and energy balances. They also calculated the net present value (NPV) after a 15-year operation.

These results suggested that the flash ironmaking process would be economical if it is operated with natural gas. Although the use of hydrogen was not economical at the 2010 price, sensitivity analyses indicated that it could become economical with the development of hydrogen economy with mass production for application as an automobile fuel or with some publicly imposed $CO_2$ penalty.

Thus, this transformative technology has a significant economic potential in addition to considerable energy saving and reduced $CO_2$ emissions relative to the current blast furnace process.

## 4. Concluding Remarks

With the immediate and increasing gravity of global warming caused by anthropogenic $CO_2$ emissions and similarly serious increasing costs for energy, the steel industry faces a dire need for developing drastically new technologies to respond to these issues. These two issues are coupled in that current energy production is largely dependent on fossil fuels that generates $CO_2$ emissions and the sequestration of $CO_2$ requires energy. The amounts of emissions and energy requirements in current ironmaking processes including BF, DR, and SR were compared.

The Flash Ironmaking Technology developed at the University of Utah to address many of the disadvantages of BF and current alternate processes was introduced. This transformative technology removes the energy intensive cokemaking and pelletization/sintering steps by using iron ore concentrate without any further treatments, which allows considerable energy saving and reduced $CO_2$ emissions. It can be adopted in large enough scales (of the order of millions of tons of iron per year) to compete with the currently available ironmaking technologies or to feed EAF operations for steelmaking. This technology is expected to have an economic advantage over the BF route when it is operated with natural gas as the reducing agent as well as a fuel. Furthermore, the technology may be able to recover iron from fine feed materials other than iron ore concentrates such as dusts, precipitates, and other sources.

**Funding:** This research was funded by the U.S. Department of Energy under Award Numbers DE-EE0005751 (2012–2018) and DE-FC36-971D13554 (2005–2007) with cost share by the American Iron and Steel Institute (AISI) and the University of Utah, and by American Iron and Steel Institute during 2008–2011.

**Acknowledgments:** I wish to thank all my undergraduate and graduate assistants plus postdoctoral associates, too many to name all here, who worked with me during the development of the Flash Ironmaking Technology at the University of Utah over the years. Funding from the U.S. Department of Energy, the American Iron, and Steel Institute (AISI), and the University of Utah is gratefully acknowledged.

**Conflicts of Interest:** The author declares no conflict of interest. The funders had no role in the design of the study; in the collection, analyses, or interpretation of data; in the writing of the manuscript, or in the decision to publish the results.

**Disclaimer:** This report was prepared as an account of work sponsored by an agency of the United States Government. Neither the United States Government nor any agency thereof, nor any of their employees, makes any warranty, express or implied, or assumes any legal liability or responsibility for the accuracy, completeness, or usefulness of any information, apparatus, product, or process disclosed, or represents that its use would not infringe privately owned rights. Reference herein to any specific commercial product, process, or service by trade name, trademark, manufacturer, or otherwise does not necessarily constitute or imply its endorsement, recommendation, or favoring by the United States Government or any agency thereof. The views and opinions of authors expressed herein do not necessarily state or reflect those of the United States Government or any agency thereof.

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
