# Peer review of "Energy Consumption and CO2 Emissions in Ironmaking and Development of a Novel Flash Technology"

_metals, doi:10.3390/met10010054_

Round 1

Reviewer 1 Report

Dear Author,

Your paper “Energy Consumption and CO2 Emissions in Ironmaking and Development of a Novel Flash Technology” discusses about new plant for treating iron ore fines for the steelmaking. The technology is the typical flash furnace used in the copper and nickel metallurgy, but here presented as a less environmental impacting steelmaking technology.

The paper is really well organized and provide essential information with enough details to be understood by non-steelmakers.

Thus, it is certainly suitable for publication in Metals journal

However, some additional information are required, mainly because the most of the salient values are for state-of-the-art steelmaking processes comparison, whereas few data are available for this FIT.

Please find in the following a list of comments/suggestions/remarks to be fulfilled before publication:

Please update the statistic by the use of the last available statistics for DRI and steel production. I attached the links below Worldwide steel production: https://www.worldsteel.org/en/dam/jcr:f9359dff-9546-4d6b-bed0-996201185b12/World+Steel+in+Figures+2018.pdf DRI: https://www.midrex.com/wp-content/uploads/Midrex_STATSbookprint_2018Final-1.pdf Table 2: Please define briefly what process is Circored Figure 2: Could the Author clarify the meaning of X? is the reacted fraction? Please specify it in the figure caption CFD paragraph is smaller if compared with the other. Some details can be added to widen this section. For instance: which software? 1D/2D/3D simulation? Which part of the furnace was simulated? I understand that all the details are in another publication, but, at least, the most salient information in terms of simulation method can be added 8. economic analysis: please add the data used for economic evaluation in form of table. Like now, the economic discussion is hard to follow and understand One information is lacking is the productivity of your FIT (in t/years) and in which form the iron is produced (liquid or solid)? What about the slag? It can be well separated? How is the amount of unreacted iron fines remaining in the slag? Were recycling possibilities of such a slag evaluated? What about the chemical and physical properties of the used fines? Which is the chemical composition of the iron ore fines suitable for this process (oxides, sulfides, carbonates)? From which source (fines from sinter strand, dust abated from the filters, under-sieve size of iron ore,…) they come? There is a limit in the available amount of Fe in the ore that can erased the profit or the productivity of the process? These information should be added also to conclusion (in a summarized form) Again, about conclusions, in the paper is never mentioned the amount of CO2 emitted by this process as well as the energy demand. It would be better to calculate the energy and CO2 input and output in the same way done for the different steelmaking route, to better highlight the improved environmental compatibility of the FIT process Can this technology compete with BF/BOF or EAF routes in terms of production volume? Or it can be installed in an integrated steelmaking plant to recover the iron fines? Please discuss these two points

Author Response

Please update the statistic by the use of the last available statistics for DRI and steel production. I attached the links below Worldwide steel production: https://www.worldsteel.org/en/dam/jcr:f9359dff-9546-4d6b-bed0-996201185b12/World+Steel+in+Figures+2018.pdf

Thank you. This new reference was used in the revision.

Table 2: Please define briefly what process is Circored.

Defined in the revision.

 Figure 2: Could the Author clarify the meaning of X? is the reacted fraction? Please specify it in the figure caption

Corrected.

CFD paragraph is smaller if compared with the other. Some details can be added to widen this section. For instance: which software? 1D/2D/3D simulation? Which part of the furnace was simulated? I understand that all the details are in another publication, but, at least, the most salient information in terms of simulation method can be added

Suggested pieces of information have been added in the revision.

economic analysis: please add the data used for economic evaluation in form of table. Like now, the economic discussion is hard to follow and understand One information is lacking is the productivity of your FIT (in t/years) and in which form the iron is produced (liquid or solid)?

Points well taken. The suggested information has been added in the revision.

What about the slag? It can be well separated? How is the amount of unreacted iron fines remaining in the slag? Were recycling possibilities of such a slag evaluated?

These factors were not evaluated in the cited study.

What about the chemical and physical properties of the used fines? Which is the chemical composition of the iron ore fines suitable for this process (oxides, sulfides, carbonates)? From which source (fines from sinter strand, dust abated from the filters, under-sieve size of iron ore,…) they come?

These pieces of information are given in the cited reference [50]

There is a limit in the available amount of Fe in the ore that can erased the profit or the productivity of the process? These information should be added also to conclusion (in a summarized form)

Not evaluated in the cited study.

Again, about conclusions, in the paper is never mentioned the amount of CO2 emitted by this process as well as the energy demand. It would be better to calculate the energy and CO2 input and output in the same way done for the different steelmaking route, to better highlight the improved environmental compatibility of the FIT process

The entire section 3.2 discusses the energy requirement of different modes of the Flash Ironmaking. Section 3.3 discusses the levels of CO2 emissions. The only way to compare different processes is to do the calculations the same way, which is done in this and cited work.

Can this technology compete with BF/BOF or EAF routes in terms of production volume? Or it can be installed in an integrated steelmaking plant to recover the iron fines? Please discuss these two points

This are valuable comments and have been addressed in the revision. Thanks.

Reviewer 2 Report

The manuscript reports energy consumption and CO2 emissions of major ironmaking processes. It is classified as a review paper and is based on a very in-depth and detailed analysis of the literature.  Its content and industrial importance make it a good fit for Metals. The article is well written and interesting from a practical point of view. I would suggest the article for publication. Before final publication, a few comments that the authors could take into account:

Fig 3, 5: the scale bar is not readable and needs to be revised.

Pg 11, Ln 388, if “T is in K”, it would be better to keep the measurement system consistent throughout the manuscript by converting Ln 385, Ln 390… from Celsius to Kelvin.

References: The reference format could be made more uniform and consistent. Some journal titles are abbreviated, some are not. Some references are with publishing year in bold font, other are not. Please revise and make sure all bibliographical details, numbering in the list of references must be consistent, complete and accurate.

Author Response

Fig 3, 5: the scale bar is not readable and needs to be revised.

In the higher resolution picture, it is clearly visible.

Pg 11, Ln 388, if “T is in K”, it would be better to keep the measurement system consistent throughout the manuscript by converting Ln 385, Ln 390… from Celsius to Kelvin.

Point well taken. Done in the revision.

References: The reference format could be made more uniform and consistent. Some journal titles are abbreviated, some are not.

When accepted abbreviation is not certain, it is best to provide the full title so that it can be edited properly.

Some references are with publishing year in bold font, other are not.

This is according to the format suggested by this journal!!

Please revise and make sure all bibliographical details, numbering in the list of references must be consistent, complete and accurate.

They are, as far as I can see.
